# How do (perceptual) distracters distract?

**Tsvetomira Dumbalska**[1]*, **Katarzyna Rudzka**[2,3], **Hannah E. Smithson**[1], **Christopher Summerfield**[1]

**1** Department of Experimental Psychology, University of Oxford, Oxford, United Kingdom, **2** Division of Biosciences, University College London, London, United Kingdom, **3** Institute of Cognitive Neuroscience, University College London, London, United Kingdom

* tsvetomira.dumbalska@psy.ox.ac.uk

## Abstract

When a target stimulus occurs in the presence of distracters, decisions are less accurate. But how exactly do distracters affect choices? Here, we explored this question using measurement of human behaviour, psychophysical reverse correlation and computational modelling. We contrasted two models: one in which targets and distracters had independent influence on choices (independent model) and one in which distracters modulated choices in a way that depended on their similarity to the target (interaction model). Across three experiments, participants were asked to make fine orientation judgments about the tilt of a target grating presented adjacent to an irrelevant distracter. We found strong evidence for the interaction model, in that decisions were more sensitive when target and distracter were consistent relative to when they were inconsistent. This consistency bias occurred in the frame of reference of the decision, that is, it operated on decision values rather than on sensory signals, and surprisingly, it was independent of spatial attention. A normalization framework, where target features are normalized by the expectation and variability of the local context, successfully captures the observed pattern of results.

## Author summary

In the real world, visual scenes usually contain many objects. As a consequence, we often have to make perceptual judgments about a specific 'target' stimulus in the presence of irrelevant 'distracter' stimuli. For instance, when hanging a picture frame, we want to discern whether it is hanging straight, ignoring the surrounding, potentially tilted frames. Laboratory experiments have shown that the presence of distracter stimuli (i.e. other picture frames) makes this type of perceptual judgment less accurate. However, the specific effect distracters have on judgments is controversial. Here, we conducted a series of experiments to compare two alternative theories of distracter influence: one in which distracters compete with targets to determine choices (independent model) and one in which distracters wield a more indirect influence on choices (interaction model). We found evidence for the latter account. Our results suggest distracters affect perceptual decisions by adjusting how sensitive decisions are to the target stimulus.

**Data Availability Statement:** All data and code to reproduce the analyses are available in the OSF repository (https://osf.io/54rf2/) for this project.

**Funding:** This research was funded by a European Research Council grant (ERC Consolidator Award:

725937) to C.S., support from the Human Brain Project (Special Grant Agreement No: 945539) to C.S., TD was supported by an Economic and Social Research Council (ESRC) doctoral studentship and a University of Oxford Scatherd Scholarship. The funders had no role in study design, data collection and analysis, decision to publish, or preparation of the manuscript.

**Competing interests:** The authors have declared that no competing interests exist.

## Introduction

Natural environments are cluttered, and so perceptual decisions are often made about stimuli that occur in the context of distracting information. For instance, when adjusting a crooked picture frame, a decision maker must focus on its offset from vertical, ignoring the tilts of surrounding paintings, as those are irrelevant to the task. In the laboratory, one way to recreate this scenario is by asking participants to judge a cued item in a multi-element array. For example, they might discriminate the tilt of a target grating that occurs in the context of one or more distracting gratings. A large literature has focussed on understanding how the cue impacts behaviour and modulates neural signals, leading to theories of how target processing is prioritised in the face of distraction [1,2]. These models have, for example, allowed for detailed predictions about how tuning functions in visual neurons vary according to whether a preferred stimulus is attended or unattended [3–6].

Whilst studying attention has been a fertile research area, we know remarkably little about the nature and consequences of distraction itself. When a (cued) target and (uncued) distracter occur together, how exactly does the *distracter* influence choices? This question lies at the intersection of several distinct literatures in psychology and neuroscience which are rarely discussed together. Firstly, it is sometimes argued that the effects of spatial cueing can be understood through the normative lens provided by Bayesian inference. The cue provides probabilistic information about which of two locations is decision-relevant; an ideal observer will combine this with noisy signals arising from target and distracter locations, weighted by their likelihoods [7,8]. In other words, the distracter and target hold *independent* sway over decisions, with the job of the observer being to weight them appropriately according to prior information. Models of this sort have sometimes tried to subsume the literature on attention within the normative framework given by decision theory [9].

A related view of distraction can also be found in a popular theoretical framework proposing that stimuli effectively compete for neural resources, with attention acting as a controller that biases processing towards percepts that are relevant for ongoing behaviour [10]. Whilst this framework does not map unambiguously onto decision theory, it implies that decision variables are derived from admixtures of independent (relevant and irrelevant) sensory signals, with their relative weighting determined by the strength of control, for example arising in prefrontal cortex [11,12]. In the extreme, target features may be swapped for those of the distracter [13–16], particularly when attention is overloaded, split or captured by distracters. Related claims include the idea that relevant and irrelevant features race independently to drive decisions [17] or contribute to the differential weighting of mutually inhibitory sources of evidence in an accumulation-to-bound model, a framework known as *decision field theory* [18]. One specific instance of this class of theory argues that when humans choose between a preferred and a less preferred item, inputs to the decision process are given by a subtractive mixture of the two stimulus values, with a higher weight given to the currently fixated item [19].

What the models above share is a commitment to the idea that target and distracter contribute independently to choices. Broadly, we can think of the decisions as driven by a process such as

$$y^{IND} = \beta_0 + \beta_1\theta_T + \beta_2\theta_D + \varepsilon \tag{1}$$

where $y^{IND}$ is the decision variable, $\theta_T$ and $\theta_D$ are features of the target and distracter respectively, $\beta_1$ and $\beta_2$ are their respective weights, and $\varepsilon$ is an error term. We can consider the application of the weights to be outside voluntary control, so that distraction is at least in part inevitable. Here, we call this the "independent" model of distraction. For the most part, assuming that accuracy is above chance, we can expect $\beta_1 > \beta_2$. Some theories posit special

downstream mechanisms that attempt to eliminate the effect of $\beta_2$ altogether, known as *distracter suppression* [20,21].

However, a different proposal is that *T* and *D* do not wield independent sway over *y* but rather

$$y^{INT} = \beta_0 + \beta_1 \theta_T + \beta_2 \cdot f(\theta_T | \theta_D) + \varepsilon \qquad (2)$$

In this equation we use the term $f(\theta_T | \theta_D)$ to refer to some unspecified process by which target and distracter interact. For example, this could be a straightforward multiplicative process, or involve rectification or discretisation operations (for example, target processing could be stronger or weaker when the distracter is similar/dissimilar or congruent/ incongruent). What this implies is that the distracter influences the decision only by virtue of how it modulates (or interacts with) the target, and so we call this the "interaction" model of distraction.

This class of model has also been popular at various times and in various guises. One well-known literature discusses target-distracter interactions in visual search tasks, where the goal is to detect a target stimulus within an array of distracters [22,23]. Naturally, the similarity between target and distracters is a major determinant of accuracy and response times [24,25]. The feature integration model of attention captures the nature of this interaction: the effect differs depending on whether target-distracter similarity is constrained to a single feature (of at least two) or encompasses the conjunction of possible stimulus features [26,27]. The number of distracters in visual search tasks has been established as a key source of distracter influence on both target detection and identification [28]. A related line of work has examined how "crowding" a peripheral target stimulus with flanking distracters impairs its identification [29,30], typically by making it appear more consistent with the crowding distracters [31]. In contrast to these literatures, in the current project we ask how a single distracter, at a known spatial location distanced from a target, can affect categorical choices made about the target.

Interaction effects of spatially adjacent stimuli are ubiquitous in early visual processing. For instance, in the extrastriate visual cortex, neuronal responses to the receptive field centre are biased by stimulation of the surround, a phenomenon known as contextual facilitation [32]. Detection thresholds for a grating are lowered if it is flanked by colinear gratings [33–36]. Here, the distracters influence the decision by virtue of their similarity to the target. Another example involving interactions in the cortical processing of visual stimuli is the tilt illusion [37,38], whereby the tilt of a central grating is perceived as repulsed away from a ring of flanking distracters. Again, the distracters do not influence the decision directly; for example, decisions about the target are *less* repulsed if the distracters are more dissimilar, contrary to the predictions of the independent model. Similar interactions have been observed for the identification of a target letter in the presence of flanking distracters [39,40]. Target-distracter interactions have been successfully modelled in normalization frameworks, where the gain of neuronal responses is calibrated to the overall stimulation [41] and the goals of the observer [42].

Explanations of this sort have also been given for decisions about economic prospects. For example, the probability of choosing a preferred item depends on value of a decoy item which is either a distracter by virtue of being unavailable or simply less preferred. Whilst there are many explanations for this phenomenon–including those that appeal to variants of the independent model [18]–a recently popular account involves divisive normalization, stating that $y \sim \theta_T / (r + \theta_T + \theta_D)$ where *r* is a regulariser. In other words, $f(\theta_T | \theta_D)$ involves divisive operation [43,44]. Indeed, it is empirically observed that the modulation strength of a decoy depends on its proximity in conceptual space to target. Relatedly, in the cognitive domain, perception of quantities (e.g. the number of African nations) or value (the price of bottle of wine) can be

anchored by irrelevant prior information [45]. However anchoring by an irrelevant quantity also seems to diminish with distance between the anchor and the target value [46], indicative of an interaction between target and distracters. Together, these effects from diverse literatures imply a view whereby distracters and targets interact. Indeed, a recent paper showed how a version of the interaction model can explain a range of distracter-mediated phenomena in perceptual, cognitive and value-based decision tasks [47].

In the current article, we set out to characterize the functional form of the effect distracters yield on perceptual choices. To address this question, we describe three studies in which human participants made a fine discrimination judgement about a target grating that is presented adjacent to an irrelevant distracter grating. Crucially, we strived to minimize the canonical early visual processing interactions by presenting gratings to opposite visual hemifields. We used computational modelling and reverse correlation analysis to try to understand how the distracter influences choices. This approach allowed us to directly compare variants of the independent and interaction models as explanations of our data. Our experiments addressed three related questions. Firstly, how does the distracter influence decisions? Secondly, does it do so at the perceptual or decision level? And thirdly, how does spatial attention mitigate (or otherwise) the effect of distraction?

## Results

On each trial of three experiments, participants (n = 68 total) viewed pairs of Gabor patches that were presented briefly within cyan and magenta circular placeholders to the left and right of fixation (**Fig 1A**). At stimulus onset, the central fixation point changed colour to match one of the two placeholders. The stimulus within that placeholder was denoted the target for that trial, and the other stimulus was the distracter. In all experiments, participants pressed a button to make a binary judgment of the orientation of the target stimulus (clockwise or counterclockwise, CW or CCW) with respect to a boundary (reference orientation) and to ignore the unprobed distracter. Fully informative feedback was offered after each choice. Grating orientations for each stimulus were sampled uniformly between -10˚ and 10˚ degrees either side of the boundary.

Exp.1 and Exp.2 differed only in the way the boundary was selected. In Exp.1, both stimuli had the same boundary (the vertical meridian). In Exp.2, the boundaries for the two stimuli differed by 90˚ (one was vertical and the other was horizontal). We orthogonalized the two decision boundaries to disentangle the perceptual and decision level information in our task. This effectively meant that, in Exp. 2, the more similar a target and a distracter stimulus were in terms of offset from the relevant boundary, the more disparate their raw orientations. Thus, if the contextual influence of the distracter occurred at the decision level, Exp. 1 and 2 should yield equivalent contextual effects, when the data is analysed in terms of offset from the relevant boundary (as we have done here). Conversely, if the distracter influence operated on the perceptual level, we should observe effects of opposite signs in the two experiments: the more consistent a target and distracter are in decision, the more inconsistent they are, by definition, in sensory space. Note that in these experiments we did not manipulate spatial attention, and each placeholder was equally likely to be cued. In a third experiment, which is otherwise similar to Exp.1, we additionally included spatial attention as a factor in our design. We describe Exp. 3 in more detail below.

### The effect of distracter signals on choice

We define the decision-relevant features of the target $\theta_T$ and the distracter $\theta_D$ as their respective angular distances to the boundary on each trial. We assumed that choice probabilities

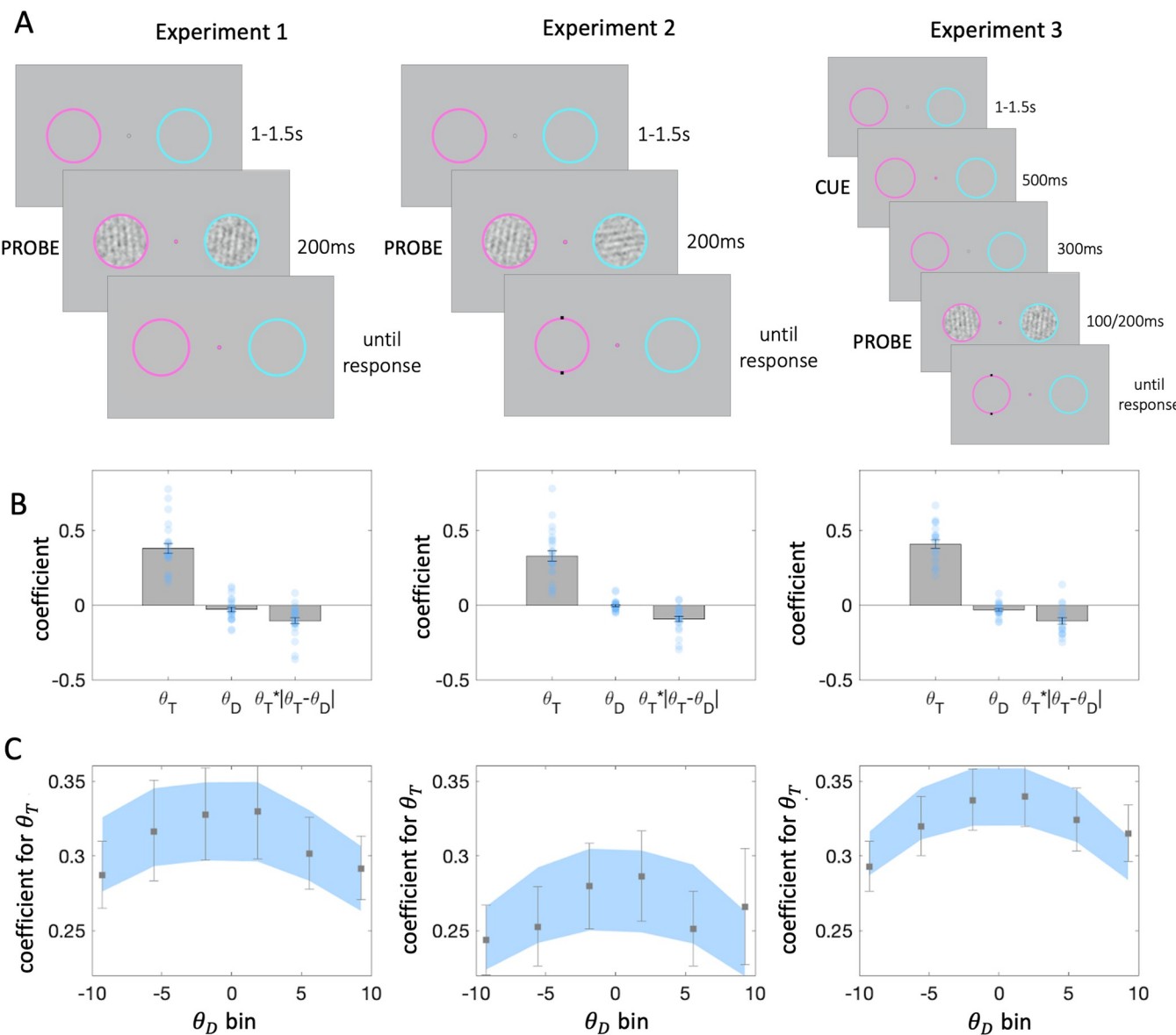

**Fig 1. A.** Experimental design. **Left**: Participants were asked to fixate on a point in the centre of the screen. Two noisy gratings appeared briefly within the coloured rings on the left and right of the fixation point. At stimulus onset, the fixation point changed colour and served as a probe. Participants had to report the tilt (CW or CCW) of the probed grating relative to vertical. **Middle**: In experiment 2, the two coloured rings had different reference orientations. In the example here, the decision boundary for the pink ring was vertical, and horizontal for the cyan ring. **Right**: In experiment 3, 800ms prior to stimulus presentation, the fixation point assumed the colour of the ring which was more likely to be probed (70% validity) for 500ms. **B.** Beta coefficients from regression of choices on target and distracter orientation. Note that the $\theta_T * |\theta_T - \theta_D|$ interaction term is divided by 10, to ensure that unit changes in each predictor variable are reported in consistent terms. Grey bars denote averages, error bars denote standard error of the mean, blue dots represent individual participants (Exp.1, n = 24; Exp.2, n = 24; Exp.3 = 20). **C.** Distracter orientation was divided in 6 equally spaced bins. Within each bin we estimated a binomial regression of choices on target orientation ($\beta_1$ only). Grey squares denote regression with human choice as the outcome variable, and error bars denote standard error of the mean. Blue region denotes standard error from the same regression, but with choice estimates derived from Eq 4 fit to human data.

were a logistic function of a decision variable, denoted $y$, computed from both target and distracter features

$$p(CW) = \frac{1}{1 + e^{-y}} \tag{3}$$

Where $p(CW)$ corresponds to the probability to respond clockwise (CW) on a given trial.

We first used a regression-based modelling approach to identify the set of predictors that best explained participants' behaviour (psychometric functions and Signal Detection Theory analyses are available in **S1 Appendix**). One challenge with the way we have framed the question above is that there is some liberty in specifying the nature of the interaction $f(\theta_T|\theta_D)$. We thus considered a number of candidates and used stepwise forward and backwards approach for including predictors (factor knock in and knock out, for a similar approach see [48]) to identify the model variant that best fit the human data (see **Methods**). The best-fitting variant was the following:

$$y = \beta_0 + \beta_1\theta_T + \beta_2\theta_D + \beta_3(\theta_T \cdot |\theta_T - \theta_D|) \qquad (4)$$

This model provided a better explanation of human choices than models that included a different functional form for $f(\theta_T|\theta_D)$. Interestingly, however, it also provided a better fit than models that omitted either the interaction term or the main effect of distraction. In other words, human data are best accounted for by models that include both an interactive and an independent term. We verified this claim statistically using Bayesian model selection [49] on cross-validated log likelihoods, observing exceedance probabilities of the Eq 4 model over a regression model including only an independent distracter effect ($\beta_2$): Exp.1, p = 0.80; Exp.2, p = 0.99; Exp.3 p ~ 1; and over a regression model including only an interaction distracter effect ($\beta_3$): Exp.1, p = 0.99; Exp. 2, p = 0.85; Exp.3 p ~ 1.

In the equation above the coefficients $\beta_1$ and $\beta_2$ encode the influence (weight) of the target and distracter on choice. The third term specifies the interaction in a form that we have previously called a "consistency bias" [50]. That means that there exists a gain control mechanism by which the influence of $\theta_T$ on choices is greater when there is higher *target-distracter consistency*, i.e. when $\theta_T$ and $\theta_D$ are more similar.

The best-fitting coefficients $\beta_1$, $\beta_2$ and $\beta_3$ are shown for all three experiments in **Fig 1B** (in Exp.3, we simply collapse over the attention factor for these initial analyses, see below). We highlight several results. Firstly, in all experiments, $\theta_T$ holds significant sway over decisions. This is expected because participants were instructed to discriminate the target item. Secondly, the effect of consistency, indexed by $\beta_3$, was reliable across all three experiments (coefficient comparison against zero, Exp1: $t_{23}$ = -5.09, $p < 0.001$, Exp2: $t_{23}$ = -5.43, $p < 0.001$, Exp3: $t_{19}$ = -5.00, $p < 0.001$). The negative sign of the coefficient means that the influence of the target on choices was greater when the target-distracter consistency was greater (i.e. a positive consistency bias). This provides support for the interaction model.

Thirdly, the effect of $\theta_D$ was somewhat erratic. T-tests conducted on $\beta_2$ at the group level yielded no effect in Exp.2 ($t_{23}$ = -0.51, $p = 0.62$) and a marginal repulsion in Exp.1 ($t_{23}$ = -1.99, $p = 0.06$). This repulsion was however significant in Exp.3 (Exp3: $t_{19}$ = -3.17, $p < 0.006$). Assuming that this is a replicable effect, the repulsion indexed by $\beta_2$ implies that (perhaps curiously) participants were more likely to respond "clockwise" if the distracter were counterclockwise and vice-versa if the distracter were clockwise, independent of the orientation of the target. We note that although coefficients were nonsignificant in Exp.1 and Exp.2 the fits of the regression model to data from individual participants suggested that the inclusion of $\theta_D$ was warranted by the variance it explained. One explanation for this is that the direction of influence of the distracter varies from individual to individual (Exp1: $\beta_2$ = -0.03 ± 0.075; Exp2: $\beta_2$ = -0.004 ± 0.037; Exp3: $\beta_2$ = -0.033 ± 0.046).

To further explore the interaction effect of the distracter, we used logistic regression to estimate the influence of $\theta_T$ on choice within each of 6 data bins defined by the value of the distracter. Sensitivity to the target was reduced for more extreme distracter orientations. This is expected from our consistency effects, because target and distracter are necessarily more

different (on average) when one of them is at the extreme. This pattern of results is also captured by the regression-predicted choices from Eq 4 (shaded blue lines in **Fig 1C**).

In Exp.2, we manipulated the decision boundaries for target and distracter to ascertain whether distraction affects sensory signals or decision level inputs. In this experiment, target and distracter stimuli that share the same tilt relative to their respective decision boundary (i.e., at the "decision" level) would be offset by 90˚ in terms of their raw tilts (i.e., at the "perceptual" level). Due to the circular nature of orientation space, the more consistent the target and distracter were in terms of decision level information (offset from the decision boundary), the less consistent they were in terms of sensory level information (raw tilt). This manipulation thus allowed us to test whether the interaction between target and distracter was observed in the decision space (i.e. in a coding format relative to the boundary) or in the naïve perceptual space in which tilt is initially coded. The effect of target-distracter consistency was very similar in Exp.1 and Exp.2. This implies that the relevant features for target and distracter interact at the decision rather than the perceptual level, i.e. at a stage in which they have been placed in a frame of reference that is relative to the decision boundary.

### Reverse correlation analysis

The analysis above describes each target and distracter with a scalar quantity that indexes the disparity between its orientation and the boundary. However, each target and distracter stimulus was in fact an image that contained information about a full range of orientations and its features can thus be described in more than one dimension. In particular, in our study the energy at each orientation varied from trial to trial for both target and distracter because of the smoothed noise that we applied. This afforded us the opportunity to use reverse correlation analysis to examine how these signal-like fluctuations in signal energy at each orientation influence choices, and thus to plot decision kernels in orientation space for both the target and distracter.

To quantify the relationship between trial-to-trial signal-like fluctuations in the noise and participant decisions, for each stimulus we first computed filter energy at each orientation (at the spatial scale of the target signals) using an ordered set of Gabor filters (**Fig 2** and Methods). Intuitively, the more the stimulus resembles a Gabor grating with a given orientation, the higher the filter energy of a noisy stimulus at that orientation. Next, we related fluctuations in energy at each orientation in each trial to participant choices via a series of parallel logistic regressions. The resulting beta coefficients captured the relationship between choice and energy at each orientation, and when plotted together constitute the *decision kernel*, i.e. estimate of the latent function mapping stimulus features on choice. The approach we describe has been used previously [51–53].

In **Fig 3B** we show the decision kernel for the target (black curve) and distracter (cyan line) stimuli averaged over all trials for each experiment. Positive kernel values indicate that an increase in energy at a given orientation is associated with higher probability of responding CW and negative kernel values with responding CCW. The peak of the target kernel is on the CW side of orientation space, close to the maximum signal-to-noise ratio of the CW and CCW energy distributions (**Fig 3A**, left panel; dashed line for Exp.1, we note that the results are consistent across all experiments; analogous plots for Exp.2 and Exp.3 are available in **S1 Appendix**). Note that the kernel tapers towards zero beyond the range of -10˚ and 10˚ degrees from which stimulus orientations were sampled. This pattern is also present in a decision kernel calculated based on the ground truth (whether the true stimulus orientation is CW/CCW) rather than participant responses (**Fig 3A**, right panel, for Exp.1).

In **Fig 3B** it can be seen that the decision kernel for the distracter was completely flat throughout orientation space. This is consistent with the fact that on its own, the distracter

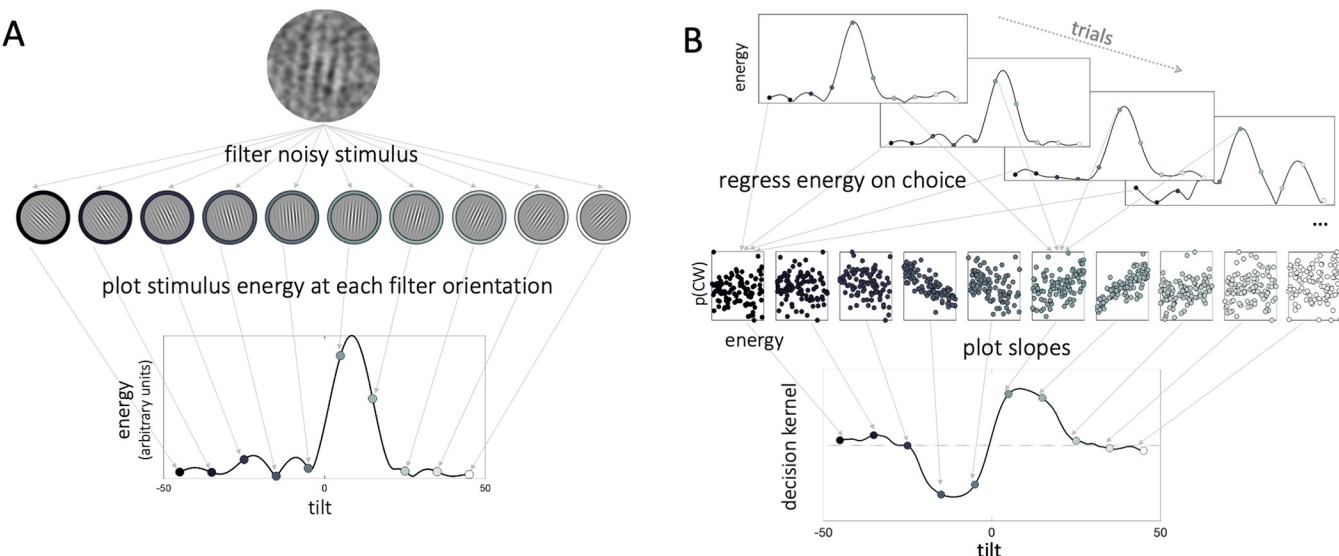

**Fig 2. Reverse correlation methods. A**: We filtered each noisy stimulus through a pool of oriented Gabor patches to calculate energy across orientation space. The higher the stimulus energy at a given orientation, the more the noisy stimulus resembles a Gabor patch of that orientation. The collection of energies of a given stimulus across orientation space constitutes the energy profile of that stimulus. **B**: We regressed choices (p(CW)) on z-scored stimulus energy via separate binomial regressions in each orientation bin. The resulting beta coefficients make up the decision kernel. Thus, the decision kernel indexes the relationship between energy at a given orientation and choice. Positive kernel values suggest that the more a stimulus resembles a Gabor patch of that orientation, the more likely is a participant to make a clockwise choice.

had a weak or inconsistent impact on choices. One approach to quantifying this intuition would be to test whether kernel values for target and distracter deviate from zero at different angles relative to the category boundary. We did conduct this analysis, and the results are shown in **S1 Appendix**. However, an issue with this approach is that the decision kernel is smooth, and so the tests conducted at each orientation are not independent. Of course, one could try to enter filter energy at each orientation into a competitive regression, but unfortunately this model is inestimable due to multicollinearity. We thus tackled this issue by using singular value decomposition (SVD) to reduce the dimensionality of our data. This approach allows us to derive a set of basis functions (components) from the data itself, and to assess their relative weighting in driving choices. The top 4 components derived from the target and distracter stimuli on each trial, which collectively accounted for 96% of the variance in filter energy (the first component captured 75% and the second– 16%), are shown in **Fig 3D**.

SVD assigns a component score to each stimulus, which tracks the variance in the stimulus explained by a given component. A negative component score means that the variance in the stimulus is best captured by a mirror image of the component, flipped around the horizontal axis. Of the 4 components, only component 2 had a different profile for the CW and CCW side of orientation space (**Fig 3E**), suggesting that the stimulus scores for the second component appear to contain information pertaining to the tilt of the stimulus (see Methods). Thus, we chose to use component 2 for analyses of participant choice, regressing participant choices on the stimulus scores for the second component for the target and distracter on each trial:

$$y = \beta_0 + \beta_1 U_T^2 + \beta_2 U_D^2 + \beta_3(U_T^2 \cdot |U_T^2 - U_D^2|) \tag{5}$$

where $U_T^2$ and $U_D^2$ denote the score of the target and distracter stimulus respectively along the second component. The coefficient $\beta_1$ reflects the independent effect of the target energy profile (as captured by component 2) on choices, $\beta_2$–the independent effect of the distracter

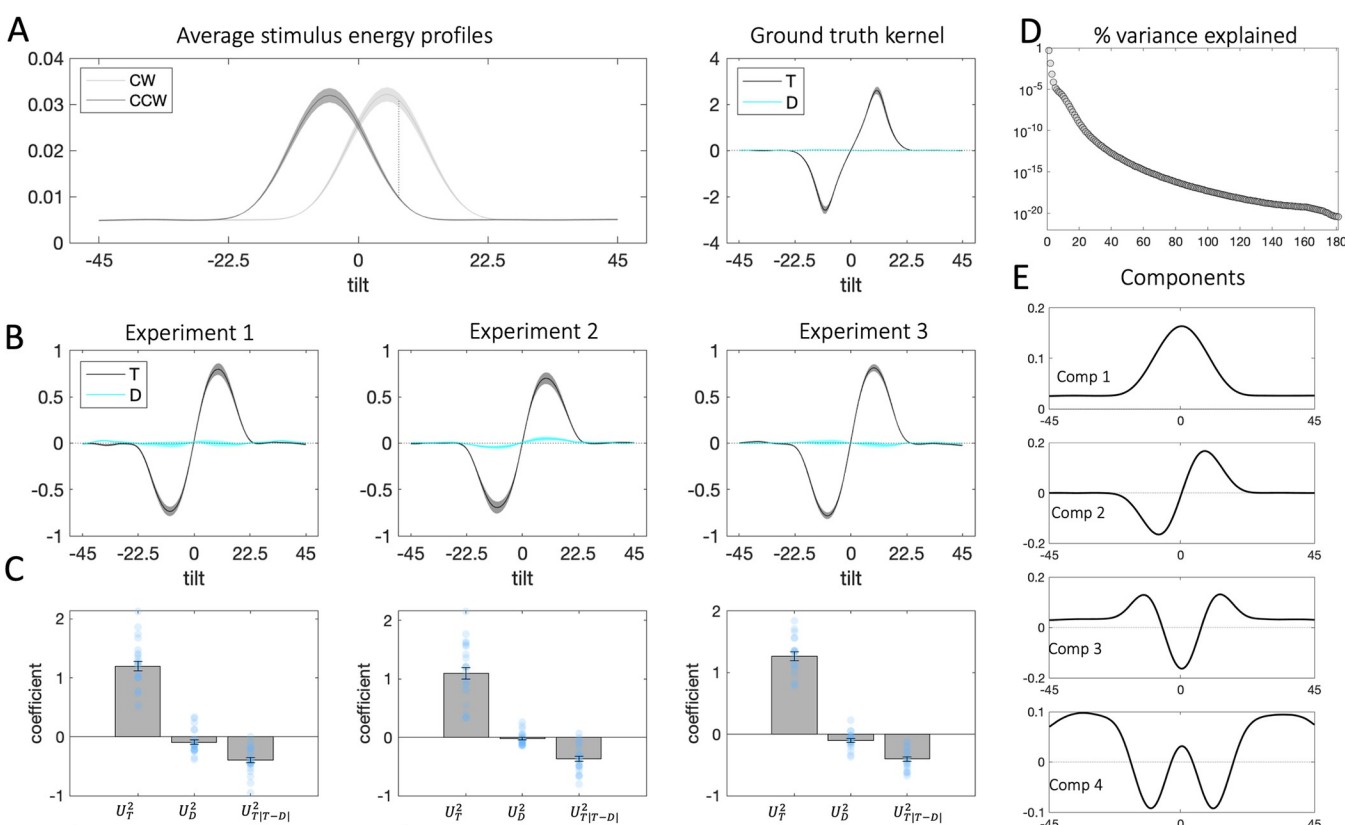

**Fig 3. A. Left**: Average stimulus energy profiles (CCW = dark grey, CW = light grey; shaded regions correspond to M±SEM; the dashed line denotes maximum signal-to-noise ratio). **Right**: Decision kernel based on ground truth stimulus tilts (target stimulus = black, distracter = cyan; shaded regions correspond to M±SEM, Exp.1). **B**: Decision kernel based on participant choices (target stimulus = black, distracter = cyan; shaded regions correspond to M ±SEM). **C**. Beta coefficients from a binomial regression of participant choices on stimulus scores for the second component and interactions with distracter signals. **D**. Percentage of variance in the stimulus energy profile explained by each of the singular value decomposition (SVD) components. Note that the y axis is in logarithmic scale. **E**. Illustration of the first four components, which together explain 96% of the variance in stimulus energy profiles.

profile, and $\beta_3$–the modulatory (interaction) effect of the distracter on sensitivity to the target, based on the consistency between target and distracter profiles. As expected, the main effect of the target profile on choices was strongly positive (Exp1: $t_{23}$ = 14.67, $p < 0.001$, Exp2: $t_{23}$ = 10.84, $p < 0.001$, Exp3: $t_{19}$ = 18.72, $p < 0.001$). In this more sensitive analysis, however, distracter profiles now exerted a small direct repulsive effect on choices in Exp. 1 and 3 (Exp1: $t_{23}$ = -2.34, $p$ = 0.03, Exp3: $t_{19}$ = -3.29, $p$ = 0.004), but not in Exp. 2 ($t_{23}$ = -1.19, $p$ = 0.24). Moreover, as expected from the analyses above, the distracter profile moderated the relationship between target profiles and choices across all experiments (Exp1: $t_{23}$ = -8.80, $p < 0.001$, Exp2: $t_{23}$ = -7.94, $p < 0.001$, Exp3: $t_{19}$ = -10.69, $p < 0.001$), such that the more consistent the target and distracter profile, the stronger the relationship between target profile and choice.

Taken together, the results from both our classic and reverse correlation analyses on Exp. 1 and Exp. 2 suggest that the effect of distraction is primarily *interactive*, i.e. distracters modulate choices by virtue of the way they interact with targets. Interestingly, this effect seemed to occur at the level of decisions, not percepts. There may also be an independent effect of the distracter on choice in Exp.1 and Exp.3, for which the category boundaries were identical for target and distracter. However, this effect was much weaker than the interactive effect, and it was *repulsive*, meaning that it pushed choices away from the orientation indicated by the distracter. We take these data to mean that the primary way in which distracters influences choices is *interactive* rather than *independent*.

## How does the effect of distraction change with spatial attention?

Focusing attention on a decision-relevant location promotes target processing at the expense of distracters, and leads to heightened accuracy and faster reaction times [1,54,55]. This naturally raises the question of how the interactive effect of distraction described here changes when targets are, or are not, the focus of spatial attention. In Exp.3, thus, we manipulated spatial attention by probabilistically cueing participants to focus on either the target or distracter stimulus in order to asses interactions with distraction. The trial sequence in Exp.3 additionally included a spatial cue that onset 800 ms before the stimuli, and signalled (with 70% validity) which of the two stimuli would be probed. We also added control blocks which consisted of trials where the cue was uninformative (50% validity), a setting equivalent to experiment 1 (except for the slight change in event timing). In the results described above, we simply collapse across the three attention conditions, yielding main effects that are very similar to those in Exp.1, as discussed in earlier sections (**Fig 1B–1C**, right panel; **Fig 3B–3C**, right panel).

To check that we had successfully manipulated attention, we first compared accuracy rates and response times across the three conditions (valid, neutral, invalid). As expected, accuracy was highest for valid trials and lowest for invalid trials, while response times followed the opposite pattern (**Fig 4A**), indicating that our attention manipulation was successful. Log-transformed response time showed a valid < neutral < invalid pattern (valid < invalid: $t_{19}$ = 8.76, p < 0.001, valid < neutral: $t_{19}$ = 4.34, p < 0.001, neutral < invalid: $t_{19}$ = 2.61, p < 0.03) and accuracy (converted to an odds ratio to avoid violating normality) showed a valid > neutral > invalid pattern (valid > invalid: $t_{19}$ = -5.54, p < 0.001, valid > neutral: $t_{19}$ = -2.73, p < 0.02, neutral > invalid: $t_{19}$ = -4.60, p < 0.001).

Next, to assess the impact of target and distracter (and their interaction) under different attentional cueing conditions we again used logistic regression (**Fig 4C**), but now including interaction terms with attention (cueing) condition:

$$y = \beta_0 + \beta_1 \theta_T + \beta_2 \theta_D + \beta_3 (\theta_T \cdot |\theta_T - \theta_D|) + \beta_4 A \cdot \theta_T + \beta_5 A \cdot \theta_D + \beta_6 A \cdot (\theta_T \cdot |\theta_T - \theta_D|) \quad (6)$$

This expression looks a little complicated but in fact builds naturally from Eq 4. The first 3 terms (associated with $\beta_1$, $\beta_2$ and $\beta_3$) are identical to those in Eq 4 above. The final 3 terms are new predictors that encode the extent to which $\theta_T$, $\theta_D$ and their interaction influence choices as a function of validity. In Eq 6 thus $A$ is an indicator variable coded as 0 on neutral trials, 1 on valid trials and -1 on invalid trials.

The results for $\beta_{1-3}$ restate those described above and so we focus on the effect of cueing. Attention modulated sensitivity to target orientations ($\beta_4$, $t_{19}$ = 4.80, $p$ < 0.001), consistent with previous reports. We can also see an attention-induced enhancement of target sensitivity in the decision kernels for the three different attention conditions (**Fig 4B**). Surprisingly, however, attention did not impact the direct effect of the distracter ($\beta_5$, $t_{19}$ = -1.46, p = 0.16), nor its modulatory influence on target sensitivity ($\beta_6$, $t_{19}$ = 0.71, p = 0.49).

We also replicated this result using reverse correlation. We defined the same regression model as in Eq 6, but with the scores for component 2 from the reverse correlation analysis ($U^2$, **Fig 4D**) as the dependent variables rather than the angular offset from the boundary $\theta$. Again, cueing condition modulated sensitivity to the target ($\beta_4$, $t_{19}$ = 5.12, $p$ < 0.001), but had no impact on the direct effect of the distracter ($\beta_5$, $t_{19}$ = -1.66, p = 0.14), nor its modulatory influence on target sensitivity ($\beta_6$, $t_{19}$ = -1.15, p = 0.27). Thus, the observed contextual effects of irrelevant information do not differ across valid, invalid and neutral trials. This implies that directing attention heightens sensitivity to the imperative (target) stimulus, but, perhaps counterintuitively, does not impact the contextual effects of irrelevant information.

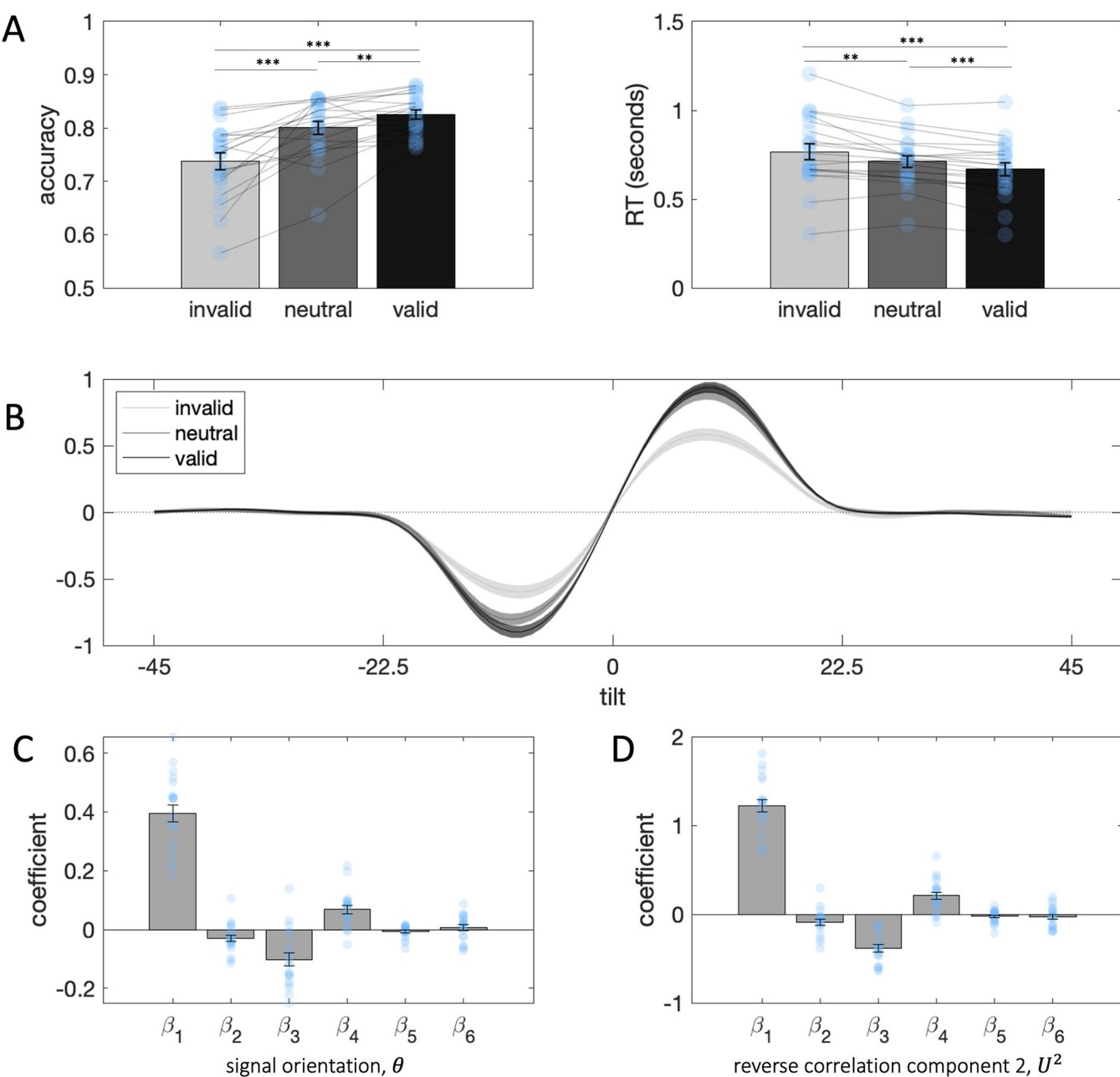

**Fig 4. A.** Manipulation checks. **Left**: Accuracy across invalid, neutral and valid trials. **Right**: Response times across invalid, neutral and valid trials; statistical tests were performed on log transformed measures. Stars denote statistical significance: $^{**}$: $p<0.05$, $^{***}$: $p<0.001$. **B.** Decision kernels for invalid, neutral and valid trials. **C.** Beta coefficients from binomial regression on signal orientation. Grey bars denote averages, error bars denote standard error of the mean, blue circles denote individual participants. $\beta_1$ is the coefficient for the target stimulus, $\beta_2$–the distracter, $\beta_3$–the interaction between target and the consistency between target and distracter, $\beta_4$–the interaction between target and cueing condition (coded as -1 = invalid, 0 = neutral, 1 = valid), $\beta_5$–the interaction between the distracter and cueing condition, $\beta_6$–the three-way interaction between target, target-distracter consistency and cueing condition. **D.** As in C, but dependent variables in the regression correspond to the score for SVD component 2 from the reverse correlation analysis.

## Normalization model

The regression model above (Eq 4) was an analytic tool designed to partition variance in choice between independent predictors ($\theta_T$ and $\theta_D$) and an interaction term $f(\theta_T|\theta_D)$. However, there is a perhaps more elegant way to model the data, and one that (although closely

related) makes more direct links to a long tradition in the cognitive sciences that appeals to neural normalization to explain attention and contextual biasing [5,41,43,44,56]:

$$y = \frac{(x - \rho \cdot \mu)}{r + \tau \cdot \sigma} \tag{7}$$

Here, a stimulus $x$ is additively/subtractively modulated by the contextual biasing term $\mu$ (in the attention literature this is sometimes known as a *contrast gain* effect, especially where the task involves detection) and multiplicatively/divisively normalized by a context variability term $\sigma$ (equivalent to a *response gain* effect); (the link between a similar version of the present model and more canonical formulations of divisive normalization is derived in [57]). The degree of each type of gain control is respectively modulated by $\rho$ and $\tau$, and $r$ is a regulariser; these 3 parameters are free to vary. Mapping this framework onto our experiment, the target feature $x = \theta_T$, the context mean $\mu = \frac{\theta_T + \theta_D}{2}$ and the context variability $\sigma = |\theta_T - \theta_D|$, i.e. it is given by the range (or dispersion) of the two stimuli.

We note that our model abstracts away from the evidence accumulation process and is agnostic in terms of the latency of the decision process. Whilst previous work has successfully modelled distracter effects in a dynamic framework, comparing distributions of response latencies of congruent and incongruent response trials [58–61], we opted to employ a parsimonious "static" model to more transparently capture the functional form of the relationship that distracters exert on choice.

This compact formulation of the model allows us to visualize how changes to context affect the transducer function linking decision inputs to choice. Changes in contextual variability adjust the slope of the transducer–the more consistent the target and distracter, the steeper the slope, and consequently, the higher the gain of processing (**Fig 5A**, left). The parameter $\tau$ controls the degree of this change. As **Fig 5B** shows, as $\tau$ grows, the interaction effect of the distracter (target-distracter consistency) increases. By contrast, the contextual expectation (the average) shifts the transducer function along the abscissa (**Fig 5A**, right). This impact translates into an independent effect of the distracter–for negative values of $\rho$ we observe an attraction towards the distracter, and for positive values of $\rho$ we observe a repulsion (**Fig 5B**, right). This rich pattern of distracter influence maps onto the variability of the independent effect observed in human participants across the three experiments (the coefficient estimates for $\beta_2$ in Eq 4).

This 3-parameter normalization model ($r$, $\tau$, and $\rho$) captures the pattern observed in the human data remarkably well (**Fig 5C**, best fitting parameter values are available in **S1 Appendix**). It recreates both the independent and interaction effect of the distracter. It is also quantitatively favoured over a model where $\tau = \rho = 0$ by Bayesian model selection on cross-validated model likelihoods (exceedance probabilities: Exp.1, p = 0.99; Exp.2, p = 0.99). To account for the observed effect of attention in experiment 3, we allowed the parameter $r$, which captures the baseline slope of the transducer above and beyond the impacts of contextual consistency, to vary freely across cueing conditions. Thus, in our process model, the spatial attention manipulation does not affect the strength of contextual normalization indexed by parameters $\tau$ and $\rho$. Rather, it changes the overall sensitivity to the target, as evidenced by the estimated parameter values: the slope of the transducer was steepest (i.e. lowest value for parameter $r$) in valid trials and shallowest (i.e. highest value for parameter $r$) in invalid trials (valid < invalid: $t_{19} = 3.94$, p < 0.001, valid < neutral: $t_{19} = 3.79$, p = 0.001, neutral < invalid: $t_{19} = 2.46$, p = 0.02). Indeed, this manipulation reproduced the observed changes in sensitivity to target orientation (**Fig 5C**, right—$\beta_4$). Moreover, it captured the attention-independent effects of the distracter ($\beta_2$ and $\beta_3$), as well as the lack of interaction of those with cueing condition ($\beta_5$ and $\beta_6$).

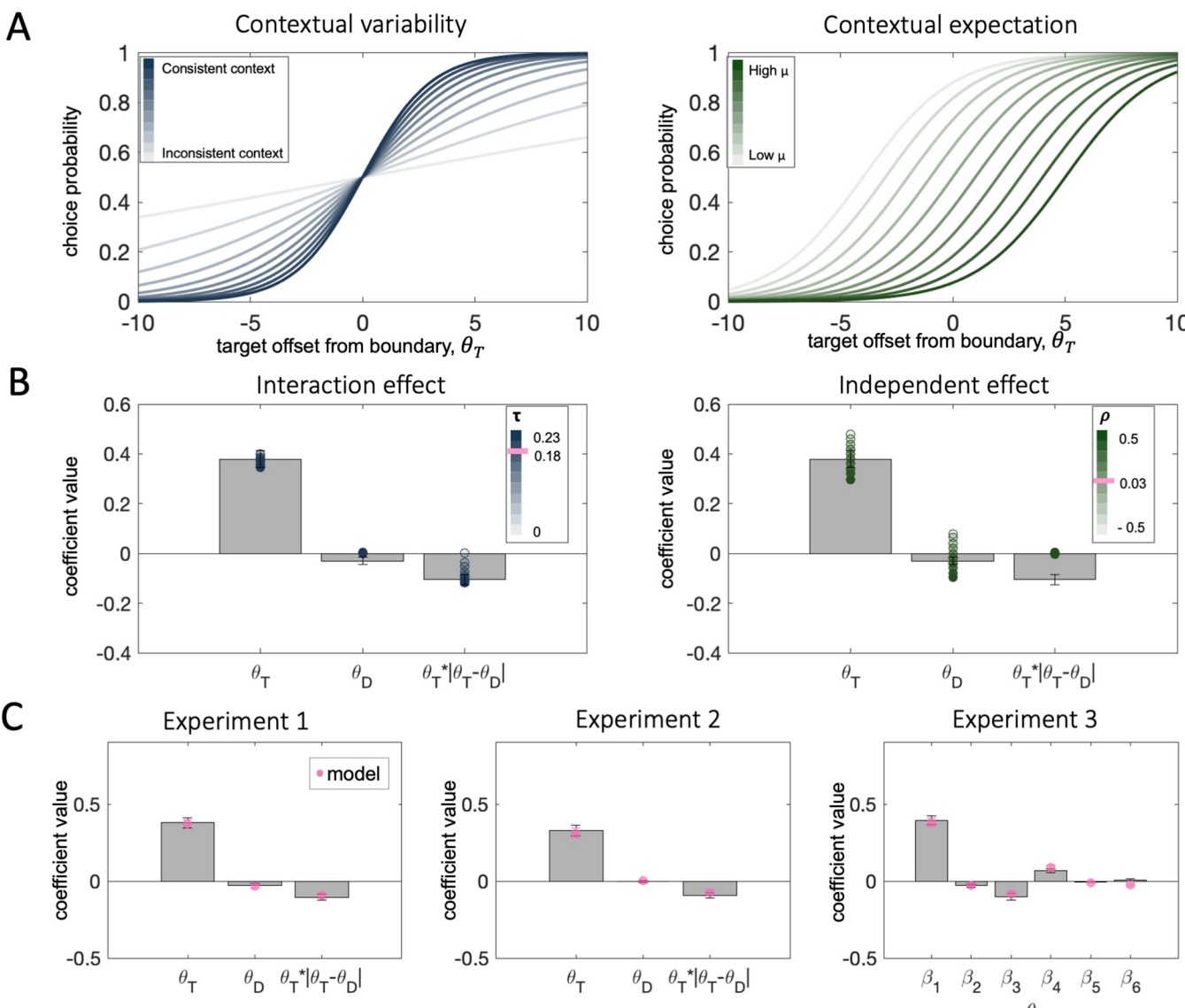

**Fig 5. A.** The effects of context on the transducer function translating decision inputs into choice probabilities. **Left**: The consistency of context changes the slope of the transducer. Lower contextual variability leads to steeper slope (darker blue curves) and vice versa for high variability (lighter blue curves). **Right:** The contextual expectation changes the location of the inflection point of the transducer. A high contextual average leads to a transducer shifted rightwards along the x-axis (darker green curves) and vice versa for low contextual average. **B.** The impact of context on the distracter effects in choice behaviour. **Left:** Contextual variability leads to an interaction effect of the distracter (target-distracter consistency). Bars correspond to coefficients from a regression on human choices from Exp.1, filled circles correspond to an analogous regression on model simulations where we varied parametrically the value of $\tau$, while keeping $\rho$ and r constant. The higher the value of the multiplicative free parameter $\tau$, the stronger the consistency effect (darker circles). The pink line depicts the average estimated value of $\tau$ for Exp. 1. **Right:** The contextual expectation leads to an independent effect of the distracter. Bars correspond to coefficients from a regression on human choices from Exp.1, filled circles correspond to an analogous regression on model simulations where we varied parametrically the value of $\rho$, while keeping $\tau$ and r constant. The higher the value of the multiplicative free parameter $\rho$, the stronger the repulsion from the distracter (darker circles). The pink line depicts the average estimated value of $\rho$ for Exp. 1. **C.** Model fits to human data. Across all three experiments bars correspond to coefficients from a regression on human choices and pink circles correspond to model-simulated choices with free parameters estimated to best fit human data. Error bars correspond to standard errors of the mean. Coefficient labels in experiment 3 are as in **Fig 4C**.

## Discussion

Perceptual decisions often occur in the presence of irrelevant information. However, the functional form of the influence distracters wield on choices remains elusive. Here we use psychophysics and modelling to address this question in the domain of perceptual decisions.

Our main result is that distracters mainly influence choices by moderating the influence that targets have on choices. Our modelling suggests that targets have more influence over choices when they are similar to distracters. This result was consistent over all three experiments using both conventional regression-based modelling and an approach based on reverse correlation analysis. This finding does not necessarily contradict, but does potentially nuance, previous accounts of distraction. For example, a standard biased competition model of attention assumes that targets and distracters compete independently to influence choice, with the job of control processes being to upweight targets at the expense of distracters. Indeed, it is easy to see how seeming support for this model might arise. For example, the omission of the interaction term in our regression model (Eq 4) can lead to spurious (positive) effect of $\beta_2$, making it look like target and distracters both wield positive influence over choices (model-simulated coefficients are available in **S1 Appendix**).

In a different study, which is notable applying a very rigorous modelling approach to a large body of data (gathered from tasks that broadly resemble our own, but including some detection tasks with a larger search array), the authors [48] characterise the effect of distracter as being a generalised decrement to precision, or an increase in overall lapses of judgment (i.e. guess rate). However, their analysis does not attempt to ask how the features of target and distracter impact choices, but rather compares experimental conditions that have different levels of distraction, which might have made it harder to detect any interaction between target and distracter that is present in that data.

Relatedly, this finding is not consistent with a familiar framework for understanding decisions based on multiple, potentially noisy, sources of information with variable decision-relevance–that which is formalised by Bayesian inference [7,8]. In our experiment, a Bayesian ideal observer would weight the target and distracter according to how relevant they were judged to be for the decision. How this relevance was determined would have to depend on some assumptions about the likely sources of error in the task, but in general this class of explanation would point to decisions being determined exclusively by a weighted mixture of independent target and distracter features. This is not what we observed.

The form of the interactive effect we did observe seems related to a phenomenon we have previously described as a *consistency bias* [50]. A consistency bias occurs when stimuli that resemble the local context are processed with heightened gain, and so have greater impact on choices. In previous work, we have reported a bias for decisions to be consistent with the recent temporal context [50] (see also [62]), which is captured by a normalization model analogous to the one we discuss here. In a closely related paper, we showed that a range of phenomena relating to spatial context–from across the subfields of perceptual, cognitive and economic decision-making–can be explained with a gain control model that resembled that described here [47]. In the previous paper, we showed how seemingly unrelated phenomena such as the tilt illusion [37], congruency effects in a Flanker task [39] and the distracting effect of decoy food items [44] can all be explained by a closely related model in which items that are similar to the context are processed with higher gain. The model employed in that paper has a slightly different functional form, as it is described in terms of the shape of neural tuning curves, but it embodies the same principle as the normalization account we describe here. In the current work, we opted for a simpler and more interpretable model to allow us to compare the independent and interactive models more transparently.

We observed an interaction between target and distracter decision values (tilt relative to the boundary) rather than perceptual values (raw tilt). This implies that the distracter, as well as the target, is placed in a frame of reference that expresses tilt relative to the category boundary. Consistent with this finding, a previous study using EEG found evidence from neural encoding that both target and distracting signals are placed in a decision-based frame of reference [63],

although of course it is likely that the extent to which selection occurs early or late depends on both the nature of the task and the available resources [64]. The data described here suggest that the interaction between target and distracter occurs at this later stage, beyond immediate sensory perception. We sought to minimize early cortical interactions in visual processing by presenting target and distracter to opposite hemifields; the nature of our task (binary categorization) likely also influenced the degree to which participants relied on (continuous) sensory representations of the stimuli. Our result is also in line with previous reports on the consistency bias[50] which also found that this effect occurred at a later stage of processing by dissociating sensory and decision inputs (via asking participants to make a decision based on the cardinality or diagonality of grating orientations). The finding that the contextual influence of perceptual distracter operates on decision-level signals could perhaps also be anticipated from the many reports of contextual effects for stimuli that share an abstract property (such as value) but not a low-level property (such as tilt), for example in the literature on decoy effects [57].

We found limited evidence that the distracter impacted choices directly. The independent effect of the distracter appeared to largely vary idiosyncratically between participants both in terms of strength and direction. On the group level, this independent effect seemed to be reliable in Exp.3 and weaker in Exp.1. Of note, these are the tasks for which the category boundary is the same for target and distracter, so it seems likely that any such effect (unlike the consistency bias) operates at a perceptual, not a decision level (some additional evidence for and discussion of this is available in S1 Appendix). It is notable that the direction of the effect which emerges at the group-level is *repulsive*, meaning that the more clockwise the distracter is, the less likely participants are to respond "clockwise". Indeed, contextual effects in sensory processing, such adaptation over time [65] and the tilt illusion [37], typically follow a repulsive pattern. Similar spatial interactions are documented in contrast perception [66,67], where a central image (e.g. texture or grating) appears of lower contrast when surrounded by a high contrast image compared to when surrounded by a uniform background. These psychophysical effects have previously been modelled in a normalization framework [41]. Here, we also demonstrate that the observed repulsive effect of the distracter can be captured by a model in which decision inputs are normalized by the contextual expectation (**Fig 5**). We note the direction of this effect cannot be explained by a model in which target and distracter compete for attention and/or processing resources. Similarly, our data did not support an account of distracter influence, where distracter features are swapped with the target (see S1 Appendix). Repulsive influences of context have also been established in the literature of serial dependencies in working memory. In those experiments, choices for the current stimulus are typically distorted towards the preceding stimulus [68,69], an attractive effect that is thought to underlie stable perception. However, the sign of this effect flips depending on the nature of the task [70,71], with tasks that require a comparison between stimuli and a standard, as our task does, documenting repulsion.

Finally, a large literature has explored the effect of attention on decisions made about a target stimulus that might be vulnerable to distraction. In Exp.3, we included a cueing manipulation that allowed us to test how attention might be deployed to mitigate the effect of distraction. Again, here, the results were quite surprising. Although we saw that spatial attention enhanced target processing–as demonstrated by both conventional and reverse correlation approaches–it did not seem to have any impact on the consistency bias. In other words, both target and distracter were elaborated into the frame of reference of the decision and then apparently interacted in a way that drove choices, but at no point (as far as we can see) did spatial attention intervene to facilitate or inhibit this process. This finding seems hard to reconcile with physiological work with non-human primates, which shows that when spatial attention is

directed towards a target stimulus, contextual modulation of neural responses by irrelevant stimuli in extra-striate area V4 is reduced by approximately 50% [72]. This discrepancy might be related to the fact that the consistency bias appears to operate on the decision level rather than on the raw sensory signals likely captured by electrophysiological work in visual areas. In our data, spatial attention appeared to multiplicatively increase the gain with which a target stimulus is encoded [41], but it did not modulate the independent or contextual effect of the distracter.

This pattern of results parallels findings on feature-based attention, which is known to operate broadly and independently of spatial attention [73], and which has been associated with a feature-similarity gain control process not dissimilar from the consistency bias reported here. For example, when attention is cued towards a visual feature—such as downwards motion—in one part of the visual field, neurons sensitive to downwards visual motion (but with receptive fields at different locations) show heightened responses [6]. In attention studies, feature gain similarity also affects performance, albeit as potentially nonmonotonic function of target-distracter feature disparity [74,75]. However, in our study there was no overt cueing of features. One possibility is that there is an automatic feature-similarity gain process that depends on the context provided by the distracter. More work would be required to understand the relationship between the phenomena reported here, and the established literature on feature-based attention.

## Methods

### Ethics statement

The study received ethical approval from the Central University Research Ethics Committee at the University of Oxford. All participants provided written informed consent.

### Experiment 1 and 2

**Participants.** Twenty-six participants (aged 25.33 ±4.38) took part in the experiment 1, and 32 (aged 24 ±4.75)–in experiment 2. Participants were compensated £10 per hour for their time. We excluded 2 participants from the analyses for experiment 1 and 8 from the analyses for experiment 2, as they performed at chance level (<55%) for one of the two stimulus locations.

Our exclusion strategy took into account that 75% performance (the staircase target and approximately the average performance in the experiment) could be achieved if participants only focused on one side of the screen (and were correct 100% for stimuli there) while guessing on the other side (50% accuracy for stimuli there). To ensure that such a strategy was not driving the observed level of accuracy, we excluded participants who performed at chance on either side of the screen. It appears that more participants (n = 8 vs n = 2) took this strategy in Exp. 2. This is perhaps indicative of the higher level of difficulty in Exp.2 due to the implementation of orthogonal decision boundaries. This change in task difficulty is also reflected in the results of the adaptive staircase (see Stimuli section below).

**Experimental procedure.** Each participant completed the experiment in a single session, lasting approximately an hour. Participants were seated in a dark room at approximately 60cm away from a computer monitor with linearized output light intensities (monitor in Exp.1: 120Hz refresh rate, 1280x1024 resolution, 21' CRT; monitor in Exp.2: 60Hz refresh rate, 1080x1290 resolution, 23' Dell LCD). Each experimental session consisted of 2 training blocks and up to 9 test blocks (100 trials each) carried out on the same day. The training blocks served to familiarize participants with the task and calibrate participant performance. To this end, during training we employed an adaptive staircase procedure (accelerated stochastic

approximation[76]) to titrate the noise contrast such that participants would perform at 75% accuracy. The staircase trial structure followed the procedure implemented in the test blocks we describe below. The only difference was that the noise contrast for the stimuli was adjusted according to the staircase, which varied the step size of the change in contrast adaptively after each response. The training blocks had a variable trial count. Each training block ended after the staircase converged and took on average between 5–10 mins. We used the contrast estimate from the second training block to generate stimuli for the test blocks.

Each test block lasted approximately 5 minutes and participants were invited to take short breaks between blocks. During the experiment, participants were asked to fixate their gaze at the centre of the screen on a fixation point between two coloured rings (magenta and cyan). The rings were located at 4° of visual angle to the left and right of centre. On each trial, after a uniformly variable interval between 1–1.5s, two noisy gratings (see stimuli for detail below) appeared within the coloured rings for 200ms. At stimulus onset, the fixation point assumed the colour of one of the two rings until the end of the trial. The colour of the fixation point served as the probe indicating which of the two stimuli was the target. Participants had to report whether the target was tilted clockwise or counterclockwise relative to a decision boundary. In experiment 1, the decision boundary was vertical for both rings (stimulus placeholders). In experiment 2, for one of the rings it was vertical, and the other–horizontal. The ring-boundary combinations were fixed for the entire duration of the experiment and counterbalanced between participants. In experiment 2, participants were reminded of the relevant decision boundary by two dots (oriented vertically or horizontally) overlayed over the target ring on each trial. Participants reported their responses via keyboard presses (left and right arrow keys) and instantaneously received fully informative auditory feedback (correct: high tone, 880Hz and incorrect: low tone, 440 Hz).

**Stimuli.** Visual stimuli were created and presented with Psychtoolbox-3 [77]. Each stimulus comprised a Gabor pattern with added smoothed Gaussian noise. Stimulus size spanned 4° visual angle and stimulus position was centred at 4° of visual angle to the left and right of fixation. The Gaussian envelope of the Gabor patterns had a standard deviation of 1° of visual angle. The phase of each stimulus was sampled from a uniform distribution. Orientation was drawn from a uniform distribution with range −10° to 10° offset from a vertical (Exp.1 and Exp.2) or horizontal boundary (Exp.2). Noise was generated (independently for each stimulus) by sampling pixel values from a Gaussian distribution and passing the resulting pixel values through a smoothing Gaussian filter. The dimension of the filter, 0.083° of visual angle, was chosen to maximize the trial-to-trial variability of the convolution between the smoothed noise and target signal. The contrast of the noise and the contrast of the target signal grating summed up to 1. We adaptively determined the contrast for the noise for each participant (see procedure for staircase details). The average value of signal contrast was 0.35 in experiment 1 and 0.42 in experiment 2.

## Experiment 3

**Participants.** Twenty participants (aged 24.3 ±3.69) took part in the experiment 3.

**Procedure.** Each participant completed the experiment in a single session, lasting approximately three hours. Participants were seated in a dark room at approximately 60cm away from a 60Hz, 1024x768 resolution, 17' LCD monitor with linearized output light intensities. Throughout the experiment, participants' eye gaze position was monitored via eye-tracking using SR Research EyeLink 1000 at a 250Hz sampling rate. Responses were not accepted on trials where the participant failed to maintain fixation (defined as gaze location further than 2° in radius from fixation). The experimenter remained in the same room as the participant to re-

calibrate the eye tracker in between experimental blocks. Auditory feedback was delivered to the participant via headphones.

Each experimental session lasted approximately three hours and consisted of 2 training mini-blocks (50 trials each), an adaptive staircase and 9 test blocks (200 trials each, with a short break midway through). The 2 training mini-blocks served to familiarize the participant with the task and the staircase followed the same procedure as in experiments 1 and 2. In the test blocks, we added a spatial cue component which provided information for which stimulus is more likely to be the target. In 2 of the test blocks, the cue was neutral. Participants were informed at the start of the neutral blocks that the cue would not be informative and the colour of the fixation point (white) reflected this. In the remaining 7 blocks, the cue assumed the colour of the ring which was more likely (70%) to be probed.

The trials followed a similar structure to Exp.1 and Exp.2. Participants were asked to fixate their gaze at the centre of the screen on a fixation point between the two coloured rings. On each trial, after a uniformly variable interval between 1–1.5s, the fixation point assumed the colour of one of the two rings (or white in the neutral condition) for 500ms. This served as the probabilistic cue, and indicated with 70% which ring would be probed. 300ms following cue offset, two noisy gratings appeared within the coloured rings for 100ms or 200ms. Stimulus timings were randomized across trials to maximise the effect of cue validity on choice. At stimulus onset, the fixation point assumed the colour of one of the two rings until the end of the trial. The colour of the fixation point served as the probe indicating which of the two stimuli was the target. At stimulus offset, two dots were overlaid over the target ring. These dots corresponded to the vertical decision boundary relative to which participants had to report the tilt of the target. Participants reported their responses via keyboard presses (left and right arrow keys) and instantaneously received fully informative auditory feedback (correct: high tone, 880Hz and incorrect: low tone, 440Hz). The stimuli were generated following the same procedure as in Experiment 1.

## Analysis

**Stepwise regression.** We used the stepwise algorithm from the Matlab Statistics and Machine Learning Toolbox to fit the stepwise logistic regressions on participant choices. The predictor variables we considered included:

- $\theta_T$–the independent effect of the target angular offset

- $\theta_D$—the independent effect of the distracter angular offset

- $|\theta_D|$–the absolute value of the distracter angular offset

- *congruency* –a binary indicator whether the orientations of target and distracter fall on the same side of the category boundary

- $\theta_T \cdot \theta_D$—the multiplicative interaction between the target and distracter angular offsets

- $\theta_T \cdot |\theta_D|$—the multiplicative interaction between the target angular offset and the absolute value of the distracter angular offsets

- $\theta_T \cdot |\theta_T - \theta_D|$—the interaction between target angular offset and the consistency between the target and the distracter angular offsets.

We ran separate models for each individual participant, as well as an aggregate model on the collapsed data for each experiment (see **S1 Appendix**). We used Bayesian model selection (from the SPM toolbox, [49]) to compare the cross-validated log likelihoods of the model identified with the stepwise regression (including $\theta_T$, $\theta_D$, and $\theta_T \cdot |\theta_T - \theta_D|$) and (1) the independent

distracter effect model (including only $\theta_T$ and $\theta_D$ as predictors), and (2) the interaction distracter effect model (including only $\theta_D$, and $\theta_T \cdot |\theta_T - \theta_D|$) as predictors).

**Stimulus energy profiles.** To estimate decision kernels, we first computed an energy profile for each stimulus (**Fig 2A**, following methods from [51,53]). We characterized the filter energy of each stimulus $S$ across orientation space using an ordered set of Gabor filters $F_{\phi_n}^\theta$. The spatial frequency of the filters matched the spatial frequency of the Gabor patterns. Filter orientation ($\theta$) ranged between -45˚ and 45˚ in increments of .5˚. To correct for differences between the phase of the stimuli (sampled uniformly between $[0,1] \cdot 2\pi$) and the filters, we specified 5 filter phases $\phi_n$ ($[0.1,0.9] \cdot 2\pi$, in steps of $0.2 \cdot 2\pi$). We calculated stimulus response to each filter $R(S|F_{\phi_n}^\theta)$ as the amount of variance in the filter that was explained by the covariance of the filter and stimulus. We then estimated the filter phase $\phi^{max}$ (which can take any value between $[0,1] \cdot 2\pi$) at which the highest response would be obtained for that orientation, by fitting the following mathematical model:

$$R(S|F_n) = E(S|\theta) \cdot cos(\varphi^n - \varphi^{max}) \tag{8}$$

where $E(S|\theta)$ corresponds to the energy of the stimulus $S$ at orientation $\theta$, and $\varphi^n$ to the phase of filter $n$ ($n$ ranges from 1 to 5). Thus, the model assumes that the relationship between filter response and filter phase follows a cosine function. We can use gradient descent to identify the value of $\phi^{max}$ that would produce the strongest filter response, as well as predict the magnitude of that response, i.e. signal energy at that orientation, $E(S|\theta)$.

Calculating the energy profiles in this manner, that is, via convolution with Gabor filters of the same parametric specification as the target signal, ensures that we are obtaining orientation energy at the spatial scale of the target. This operation is more specific than, for instance, a Fourier transform in that it takes into account the expectation for the target (and distracter) signals. Thus, the energy profile lends itself to an intuitive interpretation—the more a noisy stimulus resembles a signal grating with a given orientation, the higher the filter energy of the stimulus at that orientation. We have made code for this procedure available in the OSF data repository for the project.

**Decision kernels.** Decision kernels quantify the parametric relationship between choice and trial-to-trial energy fluctuations in the stimulus (**Fig 2B**). We estimated kernels by running a series of parallel binomial probit regressions, one per each filter orientation, with participant choice (CW or CCW) as the outcome variable and target energy as the predictor:

$$P(CW) = \varphi \left( \beta_0 + \beta_1 * z \left[ E(T_i|\theta) \right] \right) \tag{9}$$

where $\varphi(\ldots)$ refers to the probit decision rule (i.e., the cumulative normal distribution) and $z$ $(\ldots)$ to the normal deviate function (i.e. energy was standardized within each orientation bin as a z-score). In addition to the kernel quantifying the relationship between target energy and choice, we calculated kernels quantifying the direct effect of distracter energy on choice. We estimated the direct distracter effect by adding the distracter energy as a competitive regressor in the model above. The reverse correlation approach we describe has been used previously to investigate the effects of feature expectation and attention, as well as spatial attention to target stimuli on choice sensitivity [78,79,51].

**Kernel decomposition.** We carried out singular value decomposition (SVD) on a matrix containing the energy profiles of all stimuli (both targets and distracters) within an experiment. The rows of the matrix indexed individual stimuli and the columns indexed orientation. The SVD analysis produced a set of components and assigned component scores for each stimulus, which correspond to the variance in the stimulus explained by a given component.

Next, we identified the (minimum number of) components $V^i$ which explained 95% of the variance in the data and assessed which of them contained information about the tilt of the stimulus relative the decision boundary via visual inspection. We then used the component scores $U^i$ for the tilt-informative component(s) of the target and distracter stimuli as predictor variables in a regression model. The model was analogous to the regression model identified through the stepwise algorithm. The only difference was that we used the component scores for the target $U_T^i$ and distracter $U_D^i$ instead of their angular offsets, $\theta_T$ and $\theta_D$, as predictor variables.

We identified the second component for our analyses, as of the top 4 components which collectively captured 95% of the variance, this was the only component that was informative about stimulus tilt. Components 1, 3 and 4 were symmetric around the decision boundary, indicating that they tracked other, non-tilt related features of the stimuli. Further, the stimulus scores for the second component were predominantly positive for CW stimuli, and predominantly negative for CCW stimuli. A negative component score means that the variance in the stimulus is best captured by a mirror image of the component, flipped around the horizontal axis. By contrast, stimulus scores associated with components 1, 3 and 4 did not differ across CW and CCW oriented stimuli.

**Computational model fitting.** We fit the normalization model to participant choice data via gradient descent at the single trial level. Best fitting parameters were estimated for each participant individually by minimizing negative log likelihood. Model comparisons were conducted via Bayesian model selection on cross-validated model log likelihoods. We have provided code for our model fitting on the OSF repository for the project.

## Supporting information

**S1 Appendix. Supplementary analyses.**
(PDF)

## Author Contributions

**Conceptualization:** Tsvetomira Dumbalska, Hannah E. Smithson, Christopher Summerfield.

**Data curation:** Tsvetomira Dumbalska.

**Formal analysis:** Tsvetomira Dumbalska.

**Investigation:** Tsvetomira Dumbalska, Katarzyna Rudzka.

**Methodology:** Tsvetomira Dumbalska.

**Project administration:** Tsvetomira Dumbalska, Katarzyna Rudzka.

**Resources:** Hannah E. Smithson, Christopher Summerfield.

**Supervision:** Hannah E. Smithson, Christopher Summerfield.

**Writing – original draft:** Tsvetomira Dumbalska, Hannah E. Smithson, Christopher Summerfield.

**Writing – review & editing:** Tsvetomira Dumbalska, Hannah E. Smithson, Christopher Summerfield.

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
