## [Decision Letter · Decision Letter 0]

9 Aug 2022

Dear Dr Dumbalska,

Thank you very much for submitting your manuscript "How do (perceptual) distracters distract?" for consideration at PLOS Computational Biology. Apologies for the slight delay processing your submission, which is to blame on Reviewer #3, which was never found. Since I don't want to delay the process any further and since the two secured reviews are highly consistent in their judgement (and with my own), I decided to go with just two reviews.

The good news is that both reviewers were strongly positive about your manuscript. They raised a number of issues, but all of them are relatively minor. We are likely to accept this manuscript for publication, providing that you modify the manuscript according to the review recommendations.

You can find detailed reviews below. In addition to the reviews, I have three more points that I would like you to address (the first two are extremely minor):

1. In Fig. 2C, please equalize the y-axes, unless you have a good reason not to do so

2. Please consider adding a sentence or two about crowding in the introduction - I was slightly surprised to not see any reference to that literature (it's close to the tilt illusion, but not the same)

3. I was wondering what the effect 'swaps' would have on your results. Suppose that subjects accidentally swapped the target and distractor observations (i.e., respond to the distractor signal as if it were the target) on e.g. 10% of the trials. What would the estimated regression coefficients look like for such an observer? You could have a look at this by simulating a noisy observer without any other influences from the distractor other than the swaps. I think it would be good to rule out this alternative explanation for your data.

Sincerely,

Ronald van den Berg

Associate Editor

PLOS Computational Biology

Samuel Gershman

Deputy Editor

PLOS Computational Biology

[LINK]

Reviewer's Responses to Questions

**Comments to the Authors:**

Reviewer #1: The manuscript “How do (perceptual) distracters distract? “ of Dumbalska et al uses a set of 3 creative experiments the results of which are interpreted with regression models, psychophysical reverse correlation analyses and a divisive normalization computational model to better understand the question posed in the title. The authors center their work around 3 aspects of this question: 1) what is the nature of the influence (independent vs interactive) of the distractor on the perceptual decision about a target (binary categorization), 2) which information processing level does the distractor exert its influence (perceptual level vs decision level) and 3) how does spatial attention influence the answers to questions 1) and 2). Regression models are applied to address question 1, Experiments 1 and 2 together are able to address question 2 and experiment 3 uses spatial cues (neutral, invalid and valid 70% of the time) to address question 3. The reverse correlation analysis and divisive normalization model are used to confirm and further clarify these results: confirm the importance of interactive term between target value and the target to distractor difference to explain proportion correct in the binary categorization task, namely the consistency bias (in line with the author’s previous work Cheadle et al, 2014): the influence of the target on choices being higher when the target and distractor are consistent (close to each other in values, or otherwise having reduced contextual variability).

Overall, this paper represents a very nice formalization of how a distractor influences the categorization of a target, entails thorough analyses that build on each other and modeling. This paper is very well written, with intuitive explanations and nicely synthesized connections across literatures and approaches that are rarely all combined: visual search, reverse correlation, divisive normalization modeling, spatial attention. This work is important as it represents a novel and detailed application of the divisive normalization class of models (previously applied by the authors to other questions, Cheadle et al, 2014, Dumbalska et al, 2020) now to a task of visual search. I am very impressed and enthusiastic about this work and believe it will be a great paper and of high interest to the readers of PLOS Computational Biology. Before publication, I would like to ask the authors to provide some clarifications and additions. While the description of the disparate literatures covered is overall great and intuitive and very clear, at times more specific references, key terms and disambiguation are needed to help the readers. Several comments along this line are presented in the specific comments.

Specific comments.

1. Paragraph at lines 99-108. Consider including further citations: for instance, Palmer et al, 2000, Eckstein et al, 2011. I am not sure if it is entirely fair to say that the literature of visual search in general has been silent about how distractor information may influence or bias choices about the target itself – perhaps within the feature integration theory literature, yes. But is this not the case that in theory studies with identification tasks (i.e. Cameron et al, 2004) can address this question to some extent? One line of difference that I believe may help to be emphasized is the number of distractors (here 1, vs many of these studies, include much larger set sizes).

2. Paragraph at lines 124-139. Very nice connections to divisive normalization models. Please also include Navalpakkam and Itti, 2007 for a relevant approach.

3. Paragraph at lines 166-179. Could the authors explain a bit more the distinction between the distractor’s influence at the perceptual level vs decision level and the predictions of results of Exp. 1 and 2?

4. Lines 197-198 – Signal Detection theory analyses in Fig S1 and Table S1 –while they are standard, they could be elaborated more on in the methods – formula, etc/

5. Lines 200-201 – citations for the stepwise forward and backwards approach?

6. Line 219—the third term specifies the interaction in a form that we have previously called a “consistency bias”. Citation? I assume Cheadle et al, 2014 among others.

7. Line 414—computational modeling subtitle is perhaps not the most descriptive? Arguably the regression models before and psychophysical reverse correlation are also computational models, but not normative.

8. Paragraph at lines 483-49. Could the authors link this paragraph a bit better to the introduction? Not sure that it is fair to refer to the independent model by the standard model? Perhaps instead say independent model? Duncan and Humphreys’ 2009 and many other’s work on interactions between target and distractors in visual search may be related, though it is not immediately clear to me how.

9. Paragraph 495-501— different levels of distraction (meaning higher set sizes), vs in the author’s work. While Shen and Ma do not, Calder-Travis and Ma address the question of interactions between target and distractor. This is dissimilar from the interaction term in the regression models, but similar to what the tau term scales in the normalization model (what would be the min T-D difference in Calder-Travis and Ma, 2020). The search tasks are different (here binary categorization vs there detection (and localization)). I do not believe overlooking is thus quite fair to say; also do note task differences: a detection task of finding the target among distractors vs here binary categorization about the target. Also when writing tasks that broadly resemble our own - yes, they do, but somewhere may be useful to spell out binary categorization of target vs detection of target in a larger search array.

10. Lines 503-510….while this paragraph links this work (exp 3) to spatial cuing studies and Bayesian models with differential weighing, alludes to this through “variable decision-evidence”.

11. Lines 512-528—— line 513—citation — I assume it is Cheadle et al, 2014.

12. Figure 5. The connection with the consistency bias from the literature and the author’s previous work is great. Better word choice instead of static model (they already contrast in the text static models with diffusion models)? Perhaps just say parameters = 0. Why does figure 5 not show the parameters from the 3-parameter normalization model? May be helpful to also see them there.

References:

Palmer, J., Verghese, P., & Pavel, M. (2000). The psychophysics of visual search. Vision Research, 40(10- 12), 1227–1268. doi: 10.1016/s0042-6989(99)00244-8

Eckstein, M. P. (2011). Visual search: A retrospective. Journal of Vision, 11(5), 14–14. doi: 10.1167/ 11.5.14

Cameron, L. E., Tai, J., Eckstein, M., & Carrasco, M. (2004). Signal detection theory applied to three visual search tasks identification, yes/no detection and localization. Spatial Vision, 17(4), 295-325. doi: 10.1163/1568568041920212

Navalpakkam, V., & Itti, L. (2007, February). Search goal tunes visual features optimally. Neuron,

53 (4), 605–617. Retrieved from https://doi.org/10.1016/j.neuron.2007.01.018 doi: 10.1016/

j.neuron.2007.01.018

Reviewer #2: Dumbalska et al. present a series of three experiments and modeling that test the effect of distractor features on target perceptual decisions. The work targets the question of how ostensibly task-irrelevant information in the visual field impacts decisions about task-relevant targets. The primary modeling approach contrasts two conditions: one in which the effect of the target and distractor on decisions are independent and one in which they interact. The results suggest that both the distractor main effect parameter and the interaction term increase model fit but the interaction term is more robust. The interaction term is of the form such that smaller differences between the target and distractor features magnify the impact of the target feature (�T ) on decisions. The main effect was weaker but generally showed a repulsion effect. Similar conclusions were derived using a divisive normalization approach, and interestingly, the interaction between the context variability term and the target feature in the normalization model was of the same form as the interaction term in the classical models. Experiment 3 goes one step farther and adds a spatial attention manipulation using a spatial cue to indicate the target in advance of stimulus onset. Spatial attention primarily increased sensitivity to the target but had no effect on the distractor or the interaction term. Together, these results provide an elegant characterization of how target and distractor information interact during perceptual decision making.

The experimental approach is clever and the modeling provides evidence for the idea that distractors influence target decisions through an interactive process based on feature similarity. The fact that there was only weak evidence for an independent effect of the distractor was interesting and generates clear testable hypotheses for future studies of distractor processing. I think the work makes an important contribution to the literatures on perceptual decision making, attention, and distractor processing. I only have a few minor comments regarding the scope of conclusions drawn.

I was not fully convinced that the work had direct relevance to the question of whether/ how distractor features impact target processing in sensory systems. This is a minor point but at several points the authors seem to imply that the current findings suggest that distractors impact target processing at the decision stage but not sensory stage. I wondered to what degree this conclusion is confined to the current paradigm. Although the task requires precise encoding of the stimuli, the response required was only binary/ categorical. This task structure is likely to dictate (at least to some degree) sensory encoding. I therefore wondered if the task might have masked sensory processing effects of the distractor on the target. Moreover, the interaction term indicates that the impact of target features on decisions is influenced by the featural similarity of the target and distractor. This type of interaction (at least within a certain range of values) might be predicted by evidence accumulation at the sensory level – is it possible that this term could be interpreted that way?

There is psychophysical evidence of distractor feature swap and repulsion effects in working memory related to spatial attention that might be of interest (e.g., from lab of Julie Golomb, including Chen, Leber, and Golomb, 2019, JEPHPP, Dowd and Golomb, 2019, Psychological Science; but also related work on serial dependence effects in perception and memory, e.g., Fischer and Whitney, 2014, Nature Neuroscience; Bae and Luck, 2017, AP&P). The tasks in these other studies are quite different but it would be interesting if the authors could comment on the relationship between these phenomena (in working memory) along with the discussion of visual contrast effects, and the current findings regarding perceptual decisions. Are the repulsion effects based on the same principles?

**Have the authors made all data and (if applicable) computational code underlying the findings in their manuscript fully available?**

Reviewer #1: Yes

Reviewer #2: Yes

PLOS authors have the option to publish the peer review history of their article (what does this mean?). If published, this will include your full peer review and any attached files.

Reviewer #1: No

Reviewer #2: No

Figure Files:

Data Requirements:

Reproducibility:

References:

---

## [Decision Letter · Decision Letter 1]

27 Sep 2022

Dear Dr Dumbalska,

Thank you for your revised manuscript 'How do (perceptual) distracters distract?'. It was evaluated by the same two reviewers as in the first round, and by myself. All three of us are in agreement that you did an excellent job in addressing the comments raised in the first round and no new comments have come up.

Therefore, we are pleased to inform you that your manuscript has been provisionally accepted for publication in PLOS Computational Biology.

Best regards,

Ronald van den Berg

Academic Editor

PLOS Computational Biology

Samuel Gershman

Section Editor

PLOS Computational Biology

Reviewer's Responses to Questions

**Comments to the Authors:**

Reviewer #1: Thank you to the authors for addressing all my suggestions!

Reviewer #2: I thank the authors for their responses. I have no further comments.

**Have the authors made all data and (if applicable) computational code underlying the findings in their manuscript fully available?**

Reviewer #1: Yes

Reviewer #2: Yes

PLOS authors have the option to publish the peer review history of their article (what does this mean?). If published, this will include your full peer review and any attached files.

Reviewer #1: No

Reviewer #2: No

---

## [Editor Report · Acceptance letter]

10 Oct 2022

PCOMPBIOL-D-22-00720R1 

How do (perceptual) distracters distract?

Dear Dr Dumbalska,

I am pleased to inform you that your manuscript has been formally accepted for publication in PLOS Computational Biology. Your manuscript is now with our production department and you will be notified of the publication date in due course.

With kind regards,

Agnes Pap
